# Risk factors for unfavorable outcome and impact of early post-transplant infection in solid organ recipients with COVID-19: A prospective multicenter cohort study

Sonsoles Salto-Alejandre[1,2], Silvia Jiménez-Jorge[1,2], Nuria Sabé[3], Antonio Ramos-Martínez[4], Laura Linares[5], Maricela Valerio[6], Pilar Martín-Dávila[7], Mario Fernández-Ruiz[8], María Carmen Fariñas[9], Marino Blanes-Juliá[10], Elisa Vidal[11], Zaira R. Palacios-Baena[2,12], Román Hernández-Gallego[13], Jordi Carratalá[3], Jorge Calderón-Parra[4], María Ángeles Marcos[5], Patricia Muñoz[6,14], Jesús Fortún-Abete[7], José María Aguado[8], Francisco Arnaiz-Revillas[9], Rosa Blanes-Hernández[10], Julián de la Torre-Cisneros[11], Luis E. López-Cortés[2,12], Elena García de Vinuesa-Calvo[13], Clara M. Rosso[2,15], Jerónimo Pachón[2,16]*, Javier Sánchez-Céspedes[1,2], Elisa Cordero[1,2,16], on behalf of The COVIDSOT Working Team[¶]

1 Unit of Infectious Diseases, Microbiology, and Preventive Medicine, Virgen del Rocío University Hospital, Seville, Spain, 2 Institute of Biomedicine of Seville (IBiS), Virgen del Rocío University Hospital/CSIC/University of Seville, Seville, Spain, 3 Service of Infectious Diseases, Bellvitge University Hospital, Bellvitge Biomedical Research Institute (IDIBELL), University of Barcelona, Hospitalet de Llobregat, Barcelona, Spain, 4 Unit of Infectious Diseases, Microbiology, and Preventive Medicine, Puerta de Hierro University Hospital, Madrid, Spain, 5 Service of Infectious Diseases, Clinic University Hospital, Barcelona, Spain, 6 Service of Clinical Microbiology and Infectious Diseases, Sanitary Research Institute, Gregorio Marañón University Hospital, Madrid, Spain, 7 Service of Infectious Diseases, Ramón y Cajal University Hospital, Madrid, Spain, 8 Unit of Infectious Diseases, 12 de Octubre University Hospital, 12 de Octubre Hospital Research Institute (i +12), Madrid, Spain, 9 Service of Infectious Diseases, Marqués de Valdecilla University Hospital, Marqués de Valdecilla-IDIVAL, University of Cantabria, Santander, Spain, 10 Section of Infectious Diseases, La Fe University Hospital, Valencia, Spain, 11 Service of Infectious Diseases, Reina Sofía University Hospital, Maimónides Biomedical Research Institute of Córdoba (IMIBIC), Córdoba, Spain, 12 Unit of Infectious Diseases, Microbiology, and Preventive Medicine, Virgen Macarena University Hospital, Seville, Spain, 13 Unit of Kidney Transplant, Service of Nephrology, Badajoz University Hospital, Extremadura, Spain, 14 CIBERES (CB06/06/0058), Department of Medicine, Faculty of Medicine, Complutense University, Madrid, Spain, 15 Unit of Clinical Investigation and Clinical Trials, Virgen del Rocío University Hospital, Seville, Spain, 16 Department of Medicine, University of Seville, Seville, Spain

¶ Membership of the authors belonging to The COVIDSOT Working Team is listed in the Acknowledgments.
* pachon@us.es

## Abstract

The aim was to analyze the characteristics and predictors of unfavorable outcomes in solid organ transplant recipients (SOTRs) with COVID-19. We conducted a prospective observational cohort study of 210 consecutive SOTRs hospitalized with COVID-19 in 12 Spanish centers from 21 February to 6 May 2020. Data pertaining to demographics, chronic underlying diseases, transplantation features, clinical, therapeutics, and complications were collected. The primary endpoint was a composite of intensive care unit (ICU) admission and/or death. Logistic regression analyses were performed to identify the factors associated with these unfavorable outcomes. Males accounted for 148 (70.5%) patients, the median age was 63 years, and 189 (90.0%) patients had pneumonia. Common symptoms were fever,

**Data Availability Statement:** All relevant data are within the paper and its Supporting information files.

**Funding:** This study was supported by Plan Nacional de I+D+i 2013-2016 and Instituto de Salud Carlos III, Subdirección General de Redes y Centros de Investigación Cooperativa, Ministerio de Ciencia, Innovación y Universidades, Spanish Network for Research in Infectious Diseases (REIPI RD16/0016); co-financed by European Development Regional Fund "A way to achieve Europe", Operative Program Intelligence Growth 2014-2020. EC and JSC received grants from the Instituto de Salud Carlos III, Ministerio de Ciencia e Innovación, Proyectos de Investigación sobre el SARSCoV-2 y la enfermedad COVID-19 (COV20/00370; COV20/00580). JSC is a researcher belonging to the program "Nicolás Monardes" (C-0059–2018), Servicio Andaluz de Salud, Junta de Andalucía, Spain. SS-A is supported by a grant from the Instituto de Salud Carlos III, Ministerio de Ciencia e Innovación, Proyectos de Investigación sobre el SARS-CoV-2 y la enfermedad COVID-19 (COV20/00370).

**Competing interests:** The authors have declared that no competing interests exist.

cough, gastrointestinal disturbances, and dyspnea. The most used antiviral or host-targeted therapies included hydroxychloroquine 193/200 (96.5%), lopinavir/ritonavir 91/200 (45.5%), and tocilizumab 49/200 (24.5%). Thirty-seven (17.6%) patients required ICU admission, 12 (5.7%) suffered graft dysfunction, and 45 (21.4%) died. A shorter interval between transplantation and COVID-19 diagnosis had a negative impact on clinical prognosis. Four baseline features were identified as independent predictors of intensive care need or death: advanced age, high respiratory rate, lymphopenia, and elevated level of lactate dehydrogenase. In summary, this study presents comprehensive information on characteristics and complications of COVID-19 in hospitalized SOTRs and provides indicators available upon hospital admission for the identification of SOTRs at risk of critical disease or death, underlining the need for stringent preventative measures in the early post-transplant period.

## Introduction

In December 2019, the novel severe acute respiratory syndrome coronavirus 2 (SARS-CoV-2), causative agent of coronavirus disease 2019 (COVID-19), emerged in China [1]. It spread globally, becoming a public health emergency and a pandemic of historic dimensions [2]. Spain has been one of the most affected countries in the world in terms of absolute number of diagnosed cases and deaths per capita [3], causing a dramatic decline in donations and transplantation procedures per day, with mean numbers dropping from 7.2 to 1.2 and 16.1 to 2.1, respectively [4].

The clinical spectrum of COVID-19 ranges from asymptomatic disease to pneumonia, life-threatening complications, and ultimately death [5–7]. Risk factors for severe disease in the general population include older age and comorbidities [8], but the impact of chronic immunosuppression related to transplantation on COVID-19 is not well known. Despite widespread concern that COVID-19 clinical phenotypes may be more severe among solid organ transplant recipients (SOTRs) due to a poorer inflammatory response and greater organ injury, data on this population are limited to a few case series and generally small retrospective cohorts [9–25].

As hospitals around the world prepare for a rising and maintained incidence of COVID-19, important questions on the natural history of the disease, susceptibility of SOTRs, severity risk factors, and transplant specific management of antivirals and immunosuppressants remain unanswered [26]. This multicenter study aimed to shed light on said matters, presenting the clinical characteristics, treatments, and predictors of unfavorable outcomes (intensive care unit (ICU) admission and/or death) in 210 consecutively hospitalized adult SOTRs with COVID-19.

## Materials and methods

### Design and patients

We conducted a nationwide prospective observational cohort study (S1 Table for STROBE checklist) within the Spanish Network for Research in Infectious Diseases (REIPI) and the Group for the Study of Infection in Transplantation and the Immunocompromised Host (GESITRA-IC). Investigators from the 12 participating centers from different regions of Spain were asked to include all consecutive SOTR adults hospitalized with confirmed COVID-19 by real-time polymerase chain reaction (RT-PCR) assay for SARS-CoV-2 in respiratory samples,

from 21 February to 6 May 2020. The baseline was the date of hospital admission, and the follow-up censoring date was 6 June 6 2020. The study protocol was approved by the Ethics Committee of Virgen del Rocío and Virgen Macarena University Hospitals (C.I. 0842-N-20), as well as by the proper institutional review board of each participating center (individual codes are listed in the Supporting Information), and complied with the Helsinki Declaration. Written informed consent was established as a mandatory requirement for all patients.

## Data collection

The data source was the electronic medical record system. Anonymized data were collected using an electronic Case Report Form (eCRF) and added to a database specifically designed for this study built using Research Electronic Data Capture (REDCap) tools [27]. The registered variables included demographics, comorbidities, transplant type and date, signs and symptoms at admission, baseline laboratory tests and chest X-ray findings, complications during hospitalization, management of immunosuppression, therapeutics with purported activity against COVID-19, adjunctive strategies to modulate the host inflammatory response, and clinical outcomes.

## Event of interest

The clinical outcomes of patients after 30 days follow-up were categorized into favorable (full recovery and discharged or stable clinical condition) and unfavorable (admission to ICU or death). For patients who were discharged and subsequently readmitted during the study period, only the first hospital admission episode was considered for purposes of analysis. The primary endpoint was the occurrence of an unfavorable outcome, that is, a composite of ICU admission and/or death.

## Statistical approach

A descriptive analysis of all obtained data was performed. Categorical variables were presented as n (%) and continuous variable as mean (standard deviation (SD)) or median (interquartile range (IQR)) according to the normality of the distribution. We used the $\chi^2$-test, Yates' Correction for Continuity, Student's $t$-test, or Welch's $t$-test to compare between-group differences, as appropriate.

To examine factors associated with unfavorable clinical outcomes, quantitative variables were dichotomized based on normal ranges and in the cut-offs associated with unfavorable outcomes in the general population [28], after addressing their effects as continuous. Univariable and multivariable logistic regression analyses were performed, and bivariate relationships between all predictors were thoroughly explored to account for potential confounding, collinear, and interaction effects.

For obtaining a reduced set of variables from the predictors identified in the univariable analysis, a multivariable analysis was carried out using three criteria to achieve the most accurate model: relevance to clinical situation, statistical significance ($P < 0.10$), and adequate number of events to allow for meaningful analysis. An automated backward stepwise selection was used for exclusion of variables utilizing a 5% probability threshold [29]. Gender, presence of comorbidities, lung transplantation, and immunosuppression regimens with high doses of mofetil mycophenolate ($\geq$1080 mg/day) or prednisone ($\geq$20 mg/day) appeared as possible confounders and were therefore included in the final model for adjustment. White blood cell count and oxygen saturation were excluded to prevent collinearity, since neutrophil count and respiratory rate were part of the model. We found no clinically meaningful interactions among

the potential ones examined (sex and inflammatory markers, age, and immunity response), which were not therefore included in the model as a term.

Although there are no defined well-validated measures of immunosuppression intensity, we performed a univariable analysis to specifically assess the following as possible surrogates in accordance with prior studies: earlier time post-transplant, thoracic (lung or heart) compared to non-thoracic graft, receipt of augmented mofetil mycophenolate and prednisone dosages, and higher number of baseline maintenance immunosuppressive agents [12, 30, 31]. To further ascertain the impact of a shorter interval between transplantation and COVID-19 diagnosis, as well as the type of transplant received, on unfavorable outcome, we carried out a sensitivity analysis where the roles of the dependent and independent variables were inverted.

Analyses were done using the software package SPSS (Version 26.0. Armonk, NY: IBM Corp.). All *P*-values were derived from two-tailed tests, and those <0.05 were considered statistically significant.

## Results

### Patients' characteristics and clinical presentation

The cohort included 210 hospitalized adult SOTRs in which SARS-CoV-2 was detected by RT-PCR from nasopharyngeal swabs (97.6%), sputum (1.9%), and endotracheal aspirate (0.5%). One hundred eight (51.4%) patients were kidney recipients, 50 (23.8%) were liver, 33 (15.7%) were heart, 15 (7.1%) were lung, and 4 (1.9%) were kidney–pancreas recipients. The median time from transplant to COVID-19 diagnosis was 6.6 (IQR 2.8–13.1) years. Six (2.9%) patients were in the first month posttransplant, 12 (5.7%) in the first three months, 18 (8.6%) in the first six months, and 29 (13.8%) in the first-year posttransplant. The median admission date was 25 March 2020, with little variability between centers (IQR from March 18 to April 1). Median length of hospitalization was 13 (IQR 7–19) days. Sixty-three (30.0%) patients experienced an unfavorable outcome at final follow-up, and 147 (70.0%) patients had a favorable course of the disease. Patients' characteristics, of the total cohort and categorized by clinical outcome, are shown in Table 1.

In brief, males accounted for 148 (70.5%) patients, the median age was 63 (IQR 51–71) years, and 28.6% were ≥70 years old. The age distribution of patients stratified by clinical outcome is shown in Fig 1. Age ≥70 years (*P* = 0.001) and shorter time from transplantation (*P* = 0.048) were associated with a poor clinical result, unlike other baseline demographics including sex or type of graft. At least one comorbidity was present in 85.2% patients, the most common being chronic kidney disease (35.2%), followed by diabetes mellitus (33.3%) and chronic cardiopathy (25.7%), all of which were more prevalent in the unfavorable outcome group. The median duration of symptoms before hospitalization was six (IQR 3–10) days, and the most common symptoms were fever (66.7%), cough (65.2%), gastrointestinal disturbances (41.0%), and dyspnea (38.6%). Dyspnea upon presentation was associated with unfavorable outcomes (*P* < 0.001), while other initial symptoms were analogous between groups. Similarly, there were no differences among baseline immunosuppression, where triple therapy was the preferred maintenance regimen, and the subsequent clinical evolution of COVID-19.

### Chest X-ray, hemodynamic, and laboratory findings

One hundred eighty-nine (90.0%) SOTRs had abnormal chest X-ray images: 85.7% within the favorable and 100% in the unfavorable outcome groups (*P* = 0.002). Patients with unfavorable clinical outcomes had higher respiratory rate (*P* < 0.001) and lower capillary oxygen saturation (*P* = 0.03) on initial presentation than those with a favorable disease course. We also found between-group differences regarding the baseline laboratory values. In terms of blood

**Table 1. Demographics, comorbidities, clinical data, and baseline immunosuppression in all patients and by clinical outcome at final follow-up.**

| | All (n = 210) | Favorable Outcome (n = 147) | Unfavorable Outcome (n = 63) | P-value |
|---|---|---|---|---|
| **Age in years, mean (SD)** | 63 (12) | 61 (11) | 65 (7) | .01 |
| Age ≥ 70 (%) | 60 (28.6) | 32 (21.8) | 28 (46.6) | .001 |
| **Male sex (%)** | 148 (70.5) | 104 (70.7) | 44 (69.8) | .90 |
| **Organ transplant (%)** | | | | |
| Kidney | 108 (51.4) | 74 (50.3) | 34 (54.0) | .63 |
| Liver | 50 (23.8) | 37 (25.2) | 13 (20.6) | .48 |
| Heart | 33 (15.7) | 24 (16.3) | 9 (14.3) | .71 |
| Lung | 15 (7.1) | 9 (6.1) | 6 (9.5) | .56 |
| Kidney-pancreas | 4 (1.9) | 3 (2.0) | 1 (1.6) | 1.00 |
| **Years from transplant to diagnosis, median (IQR)** | 6.6 (2.8–13.1) | 7.1 (3.1–13.8) | 5.5 (1.4–11.6) | .048 |
| **Comorbidities (%)** | | | | |
| Diabetes mellitus[a] | 70 (33.3) | 42 (28.6) | 28 (44.4) | .03 |
| Chronic lung disease[b] | 42 (20.0) | 27 (18.4) | 15 (23.8) | .37 |
| Chronic cardiopathy[c] | 54 (25.7) | 31 (21.1) | 23 (36.5) | .02 |
| Chronic kidney disease[d] | 74 (35.2) | 46 (31.3) | 28 (44.4) | .07 |
| Chronic liver disease[e] | 29 (13.8) | 18 (12.2) | 11 (17.5) | .32 |
| Cancer[f] | 25 (11.9) | 15 (10.2) | 10 (15.9) | .25 |
| Morbid obesity[g] | 10 (4.8) | 9 (6.1) | 1 (1.6) | .16 |
| **Presenting symptoms (%)** | | | | |
| Fever | 140 (66.7) | 101 (68.7) | 39 (61.9) | .34 |
| Rhinorrhea | 14 (6.7) | 13 (8.8) | 1 (1.6) | .10 |
| Odynophagia | 16 (7.6) | 10 (6.8) | 6 (9.5) | .69 |
| Myalgias | 54 (25.7) | 42 (28.6) | 12 (19.0) | .15 |
| Headache | 18 (8.6) | 16 (10.9) | 2 (3.2) | .07 |
| Cough | 137 (65.2) | 94 (63.9) | 43 (68.3) | .55 |
| Expectoration | 34 (16.2) | 23 (15.6) | 11 (17.5) | .74 |
| Pleuritic chest pain | 11 (5.2) | 10 (6.8) | 1 (1.6) | .22 |
| Dyspnea | 81 (38.6) | 44 (29.9) | 37 (58.7) | < .001 |
| Diarrhea | 81 (38.6) | 59 (40.1) | 22 (34.9) | .48 |
| Vomiting | 20 (9.5) | 14 (9.5) | 6 (9.5) | 1.00 |
| Impaired consciousness | 14 (6.7) | 6 (4.1) | 8 (12.7) | .046 |
| **Days from symptoms onset to diagnosis, median (IQR)** | 6 (3–10) | 6 (3–11) | 5 (3–8) | .64 |
| **Baseline immunosuppression (%)** | | | | |
| Mofetil mycophenolate | 145 (69.0) | 101 (68.7) | 44 (69.8) | .87 |
| Azathioprine | 5 (2.4) | 4 (2.7) | 1 (1.6) | 1.00 |
| Ciclosporin | 18 (8.6) | 9 (6.1) | 9 (14.6) | .05 |
| Tacrolimus | 156 (74.3) | 110 (74.8) | 46 (73.0) | .78 |
| Sirolimus/everolimus | 49 (23.3) | 38 (25.9) | 11 (17.5) | .19 |
| Prednisone | 146 (69.5) | 97 (66.0) | 49 (77.8) | .09 |

[a]Treated with insulin or antidiabetic oral drugs, or presence of end-organ diabetes-related disease.

[b]Including chronic obstructive pulmonary disease, obstructive sleep apnea, and asthma.

[c]Including cardiac insufficiency, coronary heart disease, aortic aneurysm, and peripheral arterial disease.

[d]Mild (creatinine between 1.5–2 mg/dL) or moderate/severe (creatinine > 3 mg/dL or dialysis) renal impairment.

[e]Mild (without portal hypertension) or moderate/severe (cirrhosis, varices, encephalopathy, ascites) liver disease.

[f]Presence of an active solid or hematologic malignant neoplasm.

[g]Body mass index ≥ 40 kg/m2, or ≥ 35 kg/m2 plus experiencing obesity-related health conditions.

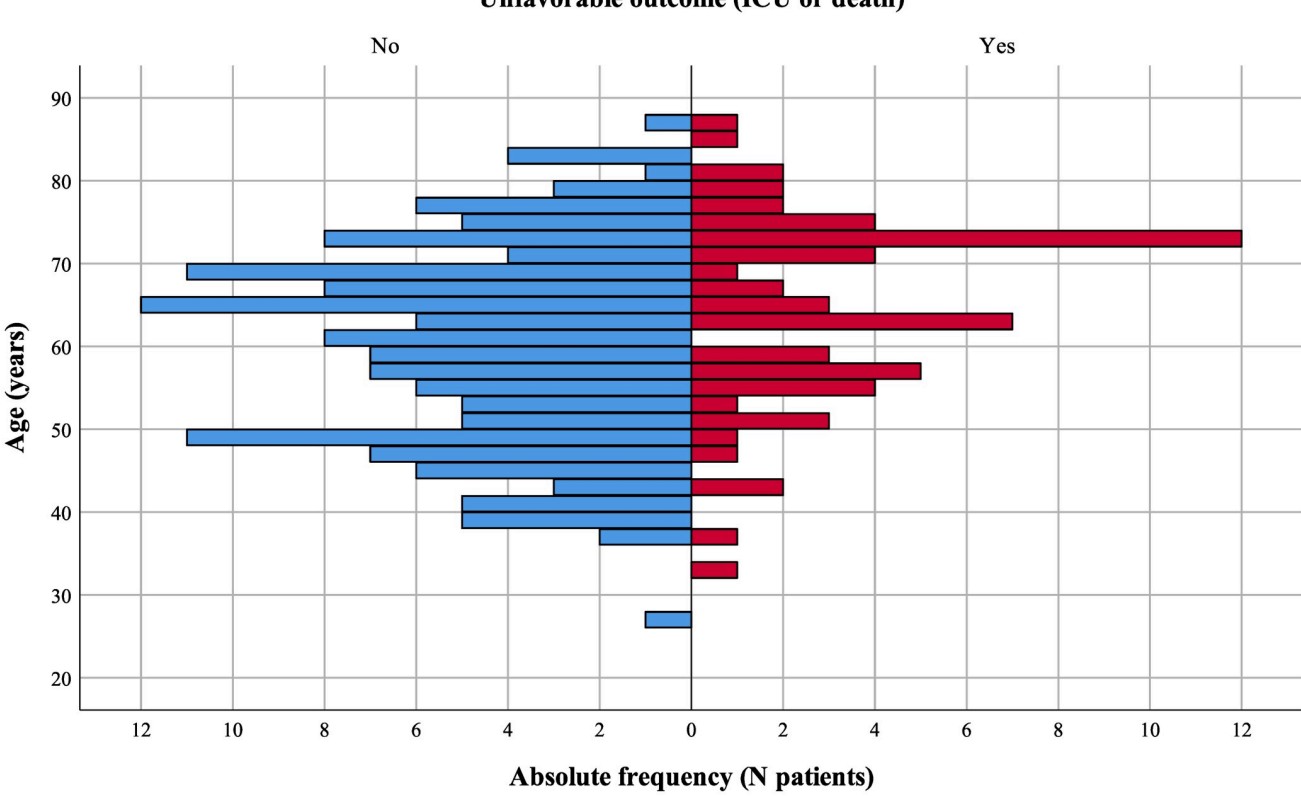

**Fig 1. Age distribution of patients stratified by clinical outcome.** Twenty-eight (46.6%) out of the 60 patients aged $\geq$ 70 years experienced an unfavorable outcome *vs.* 35 (23.3%) out of 150 patients aged < 70 years.

counts, leukocytes were higher and lymphocytes lower in the unfavorable outcome group (*P*-values, respectively, 0.04 and 0.03). By the same token, organ injury and inflammatory biomarkers such as creatinine (*P* = 0.002), lactate dehydrogenase (*P* = 0.001), C-reactive protein (*P* = 0.01), and D-dimer (*P* = 0.03) were higher among patients who later were admitted to the ICU or died. These results and additional clinical details are available in Table 2.

## Initial treatment approach, immunosuppression handling, and clinical outcomes

Antiviral or host-targeted therapies were administered to 200 (95.2%) patients, with the most used being hydroxychloroquine (193 (96.5%)), lopinavir/ritonavir (91 (45.5%)), and tocilizumab (49 (24.5%)). Lopinavir/ritonavir (*P* = 0.003) and tocilizumab (*P* < 0.001) during hospitalization, as well as high flow therapy or mechanical ventilation (*P* < 0.001), were more common practices towards severely ill patients (Table 3).

Immunosuppressive therapy was modified in 82.4% of cases, mainly by discontinuing mofetil mycophenolate and reducing tacrolimus, while maintaining prednisone dosages. For each agent, antimetabolite doses were decreased or stopped in 110/150 (73.3%) patients, calcineurin inhibitors in 119/170 (70.0%), and mTOR inhibitors in 35/49 (71.4%) patients. One hundred thirty-three out of 146 (91.1%) patients had steroid doses maintained (Table 3).

Complications were more prevalent in the unfavorable outcome group compared to the non-ICU or alive patients (*P* < 0.001). Twelve (5.7%) patients experienced graft dysfunction at

**Table 2. Initial chest x-ray imaging features, hemodynamic, and laboratory values in all patients and by clinical outcome at final follow-up in all patients and by clinical outcome at final follow-up.**

| | All (n = 210) | Favorable Outcome (n = 147) | Unfavorable Outcome (n = 63) | P-value |
|---|---|---|---|---|
| **Infiltrate on chest x-ray (%)** | 189 (90.0) | 126 (85.7) | 63 (100) | .002 |
| **Signs (%)** | | | | |
| Temperature > 37.5˚C | 59 (28.6) | 40 (27.6) | 19 (31.1) | .61 |
| Systolic blood pressure < 90 mmHg | 9 (4.5) | 8 (5.7) | 1 (1.6) | .19 |
| Diastolic blood pressure < 60 mmHg | 20 (9.9) | 13 (9.3) | 7 (11.3) | .66 |
| Hart rate > 100 bpm | 48 (25.1) | 32 (24.4) | 16 (26.7) | .74 |
| Respiratory rate > 20 bpm | 57 (31.1) | 27 (21.1) | 30 (54.5) | < .001 |
| $O_2$ sat < 95% | 61 (29.2) | 36 (24.7) | 25 (39.7) | .03 |
| **Blood counts, median (IQR)** | | | | |
| White blood cells x 1000/μL | 5.6 (4.0–7.8) | 5.3 (3.8–7.5) | 6.2 (4.4–8.2) | .04 |
| Neutrophils x 1000/μL | 4.1 (2.9–5.9) | 3.7 (2.8–5.6) | 4.7 (3.1–6.8) | .05 |
| Lymphocytes x 1000/μL | .8 (.5–1.0) | .8 (.5–1.1) | .6 (.4-.9) | .03 |
| Platelets x 1000/μL | 164 (116–214) | 158 (111–215) | 173 (123–215) | .26 |
| **Blood counts (%)** | | | | |
| White blood cells > 11 x 1000/μL | 16 (7.6) | 8 (5.4) | 8 (12.7) | .13 |
| Neutrophils > 7.5 x 1000/μL | 25 (12.3) | 14 (9.9) | 11 (17.7) | .12 |
| Lymphocytes < 1 x 1000/μL | 142 (68.6) | 94 (64.4) | 48 (78.7) | .04 |
| Platelets < 130 x 1000/μL | 67 (33.0) | 48 (33.8) | 19 (31.1) | .71 |
| **Chemistries, median (IQR)** | | | | |
| Creatinine mg/dL | 1.6 (1.1–2.3) | 1.5 (1.0–2.2) | 1.9 (1.3–2.4) | .20 |
| AST U/L | 30 (22–44) | 29 (21–42) | 37 (26–52) | .17 |
| ALT U/L | 23 (15–35) | 21 (15–32) | 27 (17–41) | 1.00 |
| Lactate dehydrogenase U/L | 270 (223–366) | 255 (207–323) | 349 (255–484) | .001 |
| **Chemistries (%)** | | | | |
| Creatinine > 1.3 mg/dL | 133 (63.9) | 83 (57.2) | 50 (79.4) | .002 |
| AST > 30 U/L | 81 (49.7) | 50 (45.5) | 31 (58.5) | .12 |
| ALT > 40 U/L | 37 (18.6) | 22 (15.9) | 15 (24.6) | .15 |
| Lactate dehydrogenase ≥ 300 U/L | 79 (40.9) | 42 (31.6) | 37 (61.7) | < .001 |
| **Additional laboratory values, median (IQR)[a]** | | | | |
| C-reactive protein mg/L | 59.6 (26.9–127.2) | 44.0 (20.6–112.6) | 89.7 (47.3–133.9) | .14 |
| D-dimer ng/mL | 612 (367–1399) | 574 (340–1060) | 799 (476–2315) | .03 |
| **Additional laboratory values (%)[a]** | | | | |
| C-reactive protein ≥ 100 mg/L | 69 (33.5) | 40 (27.8) | 29 (46.8) | .01 |
| D-dimer ≥ 600 ng/mL | 91 (52.3) | 56 (47.9) | 35 (61.4) | .09 |

[a]These values were not available for all patients (C-reactive protein N = 206, D-dimer N = 174).

the end of follow-up, resulting in transplant loss for five patients (Table 3). Overall, 37 (17.6%) SOTRs required ICU admission, and 45 (21.4%) died. A total of ten (4.8%) patients were discharged and re-admitted during the study period.

## Predictors of unfavorable outcomes

Unadjusted baseline predictors of unfavorable outcomes are shown in S2 Table. In the final multivariable analysis, adjusted for gender, comorbidities, type of transplant, and doses of immunosuppressive agents, four baseline risk factors were independently associated with increased odds of ICU admission or death: age ≥70 years (P = 0.01), respiratory rate >20 bpm

**Table 3. Treatment and complications in all patients and by clinical outcome at final follow-up.**

| | All (n = 210) | Favorable Outcome (n = 147) | Unfavorable Outcome (n = 63) | P-value |
|---|---|---|---|---|
| **Changes in immunosuppression (%)[a]** | | | | |
| Decrease or stop antimetabolite | 110/150 (73.3) | 77/105 (73.3) | 33/45 (73.3) | 1.00 |
| Decrease or stop calcineurin inhibitors | 119/170 (70.0) | 82/118 (69.5) | 37/52 (71.2) | .83 |
| Decrease or stop mTOR inhibitors | 35/49 (71.4) | 26/38 (68.4) | 9/11 (81.8) | .63 |
| Decrease or stop steroids | 13/146 (8.9) | 7/97 (7.2) | 6/49 (12.2) | .48 |
| **Viral or host-targeted medications (%)[b]** | | | | |
| Hydroxychloroquine | 193/200 (96.5) | 134/140 (95.7) | 59/60 (98.3) | .61 |
| Lopinavir/ritonavir | 91/200 (45.5) | 54/140 (38.6) | 37/60 (61.7) | .003 |
| Darunavir/cobicistat | 7/200 (3.5) | 4/140 (2.9) | 3/60 (5.0) | .74 |
| Interferon | 6/200 (3.0) | 2/140 (1.4) | 4/60 (6.7) | .12 |
| Tocilizumab | 49/200 (24.5) | 23/140 (16.4) | 26/60 (43.3) | < .001 |
| Azithromycin | 34/200 (17.0) | 28/140 (20.0) | 6/60 (10.0) | .09 |
| Methylprednisolone | 20/200 (10.0) | 14/140 (10.0) | 6/60 (10.0) | 1.00 |
| **Highest level of respiratory support (%)** | | | | |
| High flow/non-invasive mechanical ventilation | 22 (10.5) | 3 (2.0) | 19 (30.2) | < .001 |
| Intubation | 24 (11.4) | 0 (0) | 24 (38.1) | < .001 |
| **Complications during hospitalization (%)** | | | | |
| Acute respiratory distress syndrome | 54 (26.0) | 9 (6.2) | 45 (72.6) | < .001 |
| Hospital-acquired coinfections | 24 (11.9) | 9 (6.3) | 15 (25.9) | < .001 |
| Shock | 15 (7.3) | 0 (0) | 15 (25.0) | < .001 |
| Graft dysfunction | 12 (5.7) | 9 (6.1) | 3 (4.8) | .95 |
| Graft lost | 5 (2.4) | 3 (2.0) | 2 (3.2) | 1.00 |

[a]Denominator includes patients on the agent at baseline and known adjustment status.

[b]Denominator includes all patients under viral or host-targeted treatment.

($P$ = 0.001), lymphocytes <1 x 1000/μL ($P$ = 0.04), and lactate dehydrogenase ≥300 U/L ($P$ = 0.04). A forest plot presenting the respective odds ratio and 95% confidence interval is shown in Fig 2.

Among potential surrogates of immunosuppression intensity, we found a novel association between unfavorable outcomes and the temporal proximity of COVID-19 to transplantation (S3 Table). Through a series of sensitivity analyses, we further demonstrated the negative impact of an earlier post-transplant infection on clinical prognosis (Table 4), as well as the lack of association between the type of graft received and the occurrence of unfavorable outcomes (S4 Table).

Two subgroups of the study population were considered of possible higher risk: patients suffering from graft dysfunction at day 30, and those with COVID-19 acquisition during the first month post-transplant. A detailed description of their main characteristics, outcomes, and management is provided in S5 and S6 Tables.

## Discussion

In this large, prospective, nationwide study of SOTRs hospitalized with COVID-19 followed for 30 days, 17.6% required ICU admission, and the mortality rate was 21.4%. Older age, high respiratory rate, lymphopenia, and elevated level of lactate dehydrogenase at presentation were independently associated with ICU admission and/or death. Similarly, an earlier post-transplant SARS-CoV-2 infection was demonstrated as a risk factor for unfavorable outcomes.

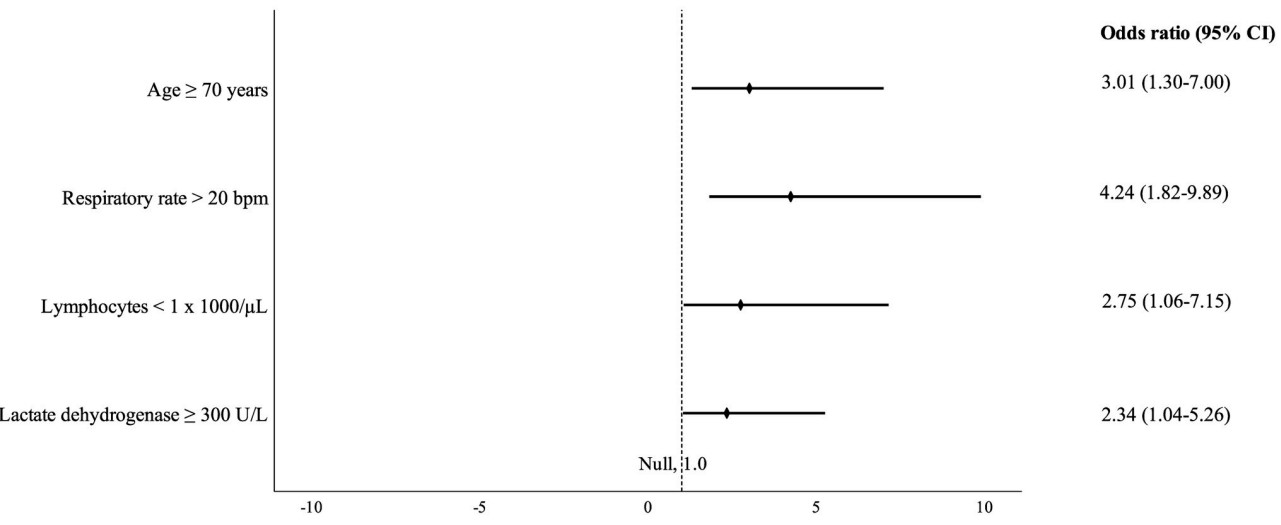

**Fig 2. Independent baseline predictors of unfavorable outcome.**

The majority of patients were male with a median age over 60 years, conforming to prior published large nationwide cohorts of the general population hospitalized with COVID-19 [32] and the 2019 Spanish National Transplant Organization Annual Report [33].

The potential negative impact of transplantation on clinical outcomes of COVID-19 has been discussed, and the few authors that directly compared results in SOTRs and general population indicated that ICU admission and death rates were higher among the immunocompromised hosts [34, 35]. However, studies including multivariable analyses of severity risk factors among hospitalized general populations with COVID-19, though with variable durations of follow-up, showed mortality and ICU admission estimates generally comparable to the ones reported for the current SOTR cohort [36–39]. The presented fatality rate in our study was also similar to the average of estimates derived from prior small and heterogeneous studies on hospitalized SOTRs [10–12, 18, 30, 34] and just one percentage point higher than the single previously published multicenter prospective SOTR cohort study (20.5%) [40]. By comparing these incidence rates with those of clinical influenza for high-risk groups, we found close resemblance in the probability of ICU admission (ranging from 11.8 to 28.6%) but less likelihood of dying (between 2.9 and 14.3%) from flu among hospitalized patients [41–43], which may be due to the existence of accessible and effective treatment.

Among the underlying comorbidities assessed, chronic cardiomyopathy, diabetes mellitus, and chronic kidney disease were all present in more than one fourth of the patients included and were associated with increased odds of unfavorable outcomes. This is in accordance with the previously described comorbidities associated with ICU admission or death in the general population [8, 36]. COVID-19 pneumonia at the time of diagnosis (defined by chest X-ray infiltrates) was also associated with unfavorable outcomes, as reported in general population studies [37, 38] and in the US multicenter SOTR cohort [40]. Moreover, no patients without pneumonia in our cohort required ICU admission or died at final follow-up, solidifying pneumonia as a major determinant of unfavorable outcomes in SOTRs.

The most common presenting symptoms in our cohort included fever, cough, and dyspnea, which were significantly associated with a poor clinical outcome. More atypical presentations, such as vomiting or diarrhea, were also reported among a significant proportion of SOTRs. This highlights that immunocompromised hosts often present with unusual or attenuated

**Table 4. Baseline risk factors, management, and outcomes *vs.* time from transplantation to COVID-19 diagnosis.**

| | All (n = 210) | ≤ 6 months from transplant to diagnosis (n = 18) | > 6 months from transplant to diagnosis (n = 192) | P-value |
|---|---|---|---|---|
| **Baseline risk factors (%)** | | | | |
| Age ≥ 70 years | 60 (28.6) | 4 (22.2) | 56 (29.2) | .53 |
| Diabetes mellitus | 70 (33.3) | 6 (33.3) | 64 (33.3) | 1.00 |
| Chronic cardiopathy | 54 (25.7) | 5 (27.8) | 49 (25.5) | 1.00 |
| Chronic kidney disease | 74 (35.2) | 5 (27.8) | 69 (35.9) | .49 |
| Dyspnea | 81 (38.6) | 8 (44.4) | 73 (38.0) | .59 |
| Respiratory rate > 20 bpm | 57 (31.1) | 7 (53.8) | 50 (29.4) | .13 |
| O$_2$ sat < 95% | 61 (29.2) | 6 (33.3) | 55 (28.8) | .69 |
| Lymphocytes < 1 x 1000/μL | 142 (68.6) | 14 (77.8) | 128 (67.7) | .38 |
| Creatinine > 1.3 mg/dL | 133 (63.9) | 12 (66.7) | 121 (63.7) | .80 |
| Lactate dehydrogenase ≥ 300 U/L | 79 (40.9) | 8 (50.0) | 71 (40.1) | .44 |
| C-reactive protein ≥ 100 mg/L | 69 (33.5) | 7 (41.2) | 62 (32.8) | .48 |
| D-dimer ≥ 600 ng/mL | 91 (52.3) | 12 (85.7) | 79 (49.4) | .05 |
| **Baseline immunosuppression (%)** | | | | |
| Mofetil mycophenolate | 145 (69.0) | 15 (83.3) | 130 (67.7) | .17 |
| Azathioprine | 5 (2.4) | 0 (0) | 5 (2.6) | 1.00 |
| Ciclosporin | 18 (8.6) | 1 (5.6) | 17 (8.9) | 1.00 |
| Tacrolimus | 156 (74.3) | 17 (94.4) | 139 (72.4) | .08 |
| Sirolimus/everolimus | 49 (23.3) | 2 (11.1) | 47 (24.5) | .32 |
| Prednisone | 146 (69.5) | 14 (77.8) | 132 (68.8) | .09 |
| **Changes in immunosuppression (%)[a]** | | | | |
| Decrease or stop antimetabolite | 110/150 (73.3) | 8/15 (53.3) | 102/135 (75.6) | .12 |
| Decrease or stop calcineurin inhibitors | 119/170 (70.0) | 9/17 (52.9) | 110/153 (71.9) | .11 |
| Decrease or stop mTOR inhibitors | 35/49 (71.4) | 2/2 (100) | 33/47 (70.2) | .91 |
| Decrease or stop steroids | 13/146 (8.9) | 2/14 (14.3) | 11/132 (8.3) | .80 |
| **Viral or host-targeted medications (%)[b]** | | | | |
| Hydroxychloroquine | 193/200 (96.5) | 15/16 (93.8) | 178/184 (96.7) | 1.00 |
| Lopinavir/ritonavir | 91/200 (45.5) | 6/16 (37.5) | 85/184 (46.2) | .50 |
| Darunavir/cobicistat | 7/200 (3.5) | 1/16 (6.3) | 6/184 (3.3) | 1.00 |
| Interferon | 6/200 (3.0) | 1/16 (6.3) | 5/184 (2.7) | .98 |
| Tocilizumab | 49/200 (24.5) | 6/16 (37.5) | 43/184 (23.4) | .34 |
| Azithromycin | 34/200 (17.0) | 0/16 (0) | 34/184 (18.5) | .11 |
| Methylprednisolone | 20/200 (10.0) | 1/16 (6.3) | 19/184 (10.3) | .86 |
| **Highest level of respiratory support (%)** | | | | |
| High flow/non-invasive mechanical ventilation | 22 (10.5) | 3 (16.7) | 19 (9.9) | .62 |
| Intubation | 24 (11.4) | 5 (27.8) | 19 (9.9) | .06 |
| **Complications during hospitalization (%)** | | | | |
| Acute respiratory distress syndrome | 54 (26.0) | 7 (41.2) | 47 (24.6) | .23 |
| Hospital-acquired coinfections | 24 (11.9) | 4 (25.0) | 20 (10.8) | .20 |
| Graft dysfunction | 12 (5.7) | 2 (11.1) | 10 (5.2) | .62 |
| Graft lost | 5 (2.4) | 1 (5.6) | 4 (2.1) | .91 |
| **Final outcome (%)** | | | | |
| Intensive care unit admission | 37 (17.6) | 8 (44.4) | 29 (15.1) | .01 |

*(Continued)*

**Table 4.** (Continued)

|  | All (n = 210) | ≤ 6 months from transplant to diagnosis (n = 18) | > 6 months from transplant to diagnosis (n = 192) | P-value |
|---|---|---|---|---|
| Death | 45 (21.4) | 6 (33.3) | 39 (20.3) | .32 |
| Unfavorable* | 63 (30.0) | 10 (55.6) | 53 (27.6) | .01 |

[a]Denominator includes patients on the agent at baseline and known adjustment status.

[b]Denominator includes all patients under viral or host-targeted treatment.

*Clinical outcome is categorized into favorable (full recovery and discharged or stable clinical condition) and unfavorable (admission to ICU or death).

signs and symptoms of infection, leading to late presentations or missed diagnosis, and potentially worse results.

Among the inflammatory parameters measured at hospital admission, creatinine, lactate dehydrogenase, C-reactive protein, and D-dimer levels were higher within the unfavorable outcome group. However, the overall variation in these biomarkers was less pronounced than that observed in the general population of hospitalized patients with COVID-19 [31, 44–46], which is biologically plausible. This being the case, further investigation is required to address whether the lower inflammatory response and greater immunosuppression characterizing SOTRs have impacts on COVID-19 clinical outcomes.

The fundamental implication of our study is the identification of specific and independent predictors (age ≥70 years, respiratory rate >20 bpm, lymphocytes <1 x 1000/μL, and lactate dehydrogenase ≥300 U/L) for unfavorable outcomes in hospitalized SOTRs with COVID-19, which could ease the development of future research and guidelines targeted at high-risk transplanted populations. Furthermore, we showed that an interval shorter than six months between transplantation and COVID-19 diagnosis has a negative impact on mortality and ICU admission rates, which is a risk that should be considered when deciding which patients should proceed with transplantation. Finally, although analogous to the general population, mortality in SOTRs hospitalized with SARS-CoV-2 infection is dramatically high, and the promotion of preventive strategies and treatments will be crucial to mitigate the adverse impacts of the COVID-19 pandemic in these patients.

The strengths of the present study are the strong design, the multicenter participation approach to make the results generalizable and comparable, the standardized and anonymous collection of data using an electronic Case Report Form, and the 30-day duration of follow-up. In parallel, we have faced some limitations. First, our study is centered on hospitalized patients, and thus the conclusions reached may not be applicable to those SOTRs attended in the outpatient setting. Second, testing limitations probably led to undercounting of mild or asymptomatic cases, and the ensuing selection bias towards more severely ill patients. Finally, the cases included only represent the early COVID-19 epidemic. Therefore, the potential benefit of therapies that are now implemented more widely, such as remdesivir and convalescent plasma, have not been addressed.

In summary, among hospitalized SOTR with COVID-19, ICU admission and death rates were high, and they were similar to those reported in the general population. Unfavorable outcomes were mainly driven by respiratory pathology (represented by a high breathing rate), older age, and two laboratory features at presentation, namely lymphopenia and elevated level of lactate dehydrogenase. An earlier post-transplant SARS-CoV-2 infection was established as a novel risk factor for ICU need and mortality. While this study provides preliminary indicators available upon hospital admission for identifying patients at risk of critical disease or death, it is an urgent priority to find efficacious antiviral treatments and to investigate the role

of the immune response in COVID-19, especially in the population of SOTRs, where it is vital to guide suitable and prompt immunomodulatory management.

## Supporting information

**S1 File. The COVIDSOT working team.**
(DOCX)

**S2 File. Institutional review board approval number of each participating center.**
(DOCX)

**S1 Table. STROBE checklist.**
(DOCX)

**S2 Table. Univariable models of baseline risk factors associated with unfavorable outcome.**
(DOCX)

**S3 Table. Univariable models of potential surrogates of immunosuppression intensity *vs*. unfavorable outcome.**
(DOCX)

**S4 Table. Clinical outcomes according to the type of transplant received.**
(DOCX)

**S5 Table. Description of patients suffering from graft dysfunction at day 30 (n = 12).**
(DOCX)

**S6 Table. Description of patients with COVID-19 acquisition during the first month post-transplant (n = 6).**
(DOCX)

**S1 Dataset. Minimal anonymized data set necessary to replicate the study findings.**
(XLSX)

## Acknowledgments

### COVIDSOT working team

*Lead author*: Elisa Cordero (elisacorderom@gmail.com).

 *Virgen del Rocío University Hospital-IBiS*, *University of Seville*, *Seville*, *Spain*: Elisa Cordero, Jerónimo Pachón, Manuela Aguilar-Guisado, Judith Berastegui-Cabrera, Gabriel Bernal-Blanco, Pedro Camacho, Marta Carretero, José Miguel Cisneros, Juan Carlos Crespo, Miguel Angel Gómez-Bravo, Carmen Infante-Domínguez, Silvia Jiménez-Jorge, Laura Merino, Clara Rosso, Sonsoles Salto-Alejandre, Javier Sánchez-Céspedes, José Manuel Sobrino-Márquez. *Bellvitge University Hospital*, *Bellvitge Biomedical Research Institute-IDIBELL*, *University of Barcelona*, *Barcelona*, *Spain*: Jordi Carratalá, Nuria Sabé, Carme Baliellas, Oriol Bestard, Carles Diez, José Gonzàlez-Costello, Laura Lladó, Eduardo Melilli. *Puerta de Hierro University Hospital*, *Madrid*, *Spain*: Antonio Ramos-Martínez, Jorge Calderón-Parra, Ana Arias-Milla, Gustavo Centeno-Soto, Manuel Gómez-Bueno, Rosalía Laporta-Hernández, Alejandro Muñoz-Serrano, Beatriz Sánchez-Sobrino. *Clinic University Hospital-IDIBAPS*, *University of Barcelona*, *Barcelona*, *Spain*: Asunción Moreno, María Ángeles Marcos, Laura Linares, Marta Bodro, María Ángeles Castel, Frederic Cofán, Jordi Colmenero, Fritz Dieckmann, Javier Fernández, Dra. Marta Farrero, Miquel Navasa, Félix Pérez-Villa, Pedro Ventura. *Gregorio Marañón*

*University Hospital*, *CIBERES*, *Madrid*, *Spain*: Maricela Valerio, Patricia Muñoz, Víctor Fernández-Alonso, Maria Olmedo-Samperio, Carlos Ortíz, Sara Rodríguez-Fernández, Maria Luisa Rodríguez-Ferrero, Magdalena Salcedo, Eduardo Zataraín. *Ramón y Cajal University Hospital*, *Madrid*, *Spain*: Pilar Martín-Dávila, Jesús Fortún-Abete, Juan Carlos Galán, Cristina Galeano-Álvarez, Francesca Gioia, Javier Graus, Sara Jiménez, Mario J. Rodríguez. *12 de Octubre University Hospital/i+12*, *CIBERCV*, *Madrid*, *Spain*: José María Aguado, Mario Fernández-Ruiz, Amado Andrés, Juan F. Delgado, Carmelo Loinaz, Francisco López-Medrano, Rafael San Juan. *Marqués de Valdecilla University Hospital-IDIVAL*, *University of Cantabria*, *Santander*, *Spain*: Carmen Fariñas, Francisco Arnaiz de las Revillas, Marta Fernández-Martínez, Ignacio Fortea-Ormaechea, Aritz Gil-Ongay, Mónica Gozalo-Marguello, Claudia González-Rico, Milagros Heras-Vicario, Víctor Mora-Cuesta. *La Fe University Hospital*, *Valencia*, *Spain*: Marino Blanes-Julia, Rosa Blanes-Hernández, Luis Almener-Bonet, María Isabel Beneyto-Castelló, Victoria Miguel Salavert-Lletí, Aguilera-Sancho-Tello, Amparo Solé-Jover. *Reina Sofía University Hospital-IMIBIC*, *Córdoba*, *Spain*: Elisa Vidal, Julián de la Torre-Cisneros, Rafael León, Álvaro Torres de Rueda, José Manuel Vaquero. *Virgen Macarena University Hospital-IBiS*, *Seville*, *Spain*: Zaira R. Palacios-Baena, Luis E. López-Cortés, David Gutiérrez-Campos, Marie-Alix Clement, Marta Fernández-Regaña, Inmaculada López-Hernández, Natalia Maldonado-Lizarazo, Ana Belén Martín-Gutiérrez, Rocío Valverde. *Badajoz University Hospital*, *Extremadura*, *Spain*: Román Hernández-Gallego, Elena García de Vinuesa-Calvo.

## Author Contributions

**Conceptualization:** Jerónimo Pachón, Javier Sánchez-Céspedes, Elisa Cordero.

**Data curation:** Silvia Jiménez-Jorge, Nuria Sabé, Antonio Ramos-Martínez, Laura Linares, Maricela Valerio, Pilar Martín-Dávila, Mario Fernández-Ruiz, María Carmen Fariñas, Marino Blanes-Juliá, Elisa Vidal, Zaira R. Palacios-Baena, Román Hernández-Gallego, Jordi Carratalá, Jorge Calderón-Parra, María Ángeles Marcos, Patricia Muñoz, Jesús Fortún-Abete, José María Aguado, Francisco Arnaiz-Revillas, Rosa Blanes-Hernández, Julián de la Torre-Cisneros, Luis E. López-Cortés, Elena García de Vinuesa-Calvo, Clara M. Rosso.

**Formal analysis:** Sonsoles Salto-Alejandre.

**Funding acquisition:** Elisa Cordero.

**Methodology:** Jerónimo Pachón, Javier Sánchez-Céspedes, Elisa Cordero.

**Project administration:** Silvia Jiménez-Jorge, Jerónimo Pachón, Elisa Cordero.

**Supervision:** Jerónimo Pachón, Javier Sánchez-Céspedes, Elisa Cordero.

**Validation:** Jerónimo Pachón, Javier Sánchez-Céspedes, Elisa Cordero.

**Visualization:** Silvia Jiménez-Jorge, Nuria Sabé, Antonio Ramos-Martínez, Laura Linares, Maricela Valerio, Pilar Martín-Dávila, Mario Fernández-Ruiz, María Carmen Fariñas, Marino Blanes-Juliá, Elisa Vidal, Zaira R. Palacios-Baena, Román Hernández-Gallego, Jordi Carratalá, Jorge Calderón-Parra, María Ángeles Marcos, Patricia Muñoz, Jesús Fortún-Abete, José María Aguado, Francisco Arnaiz-Revillas, Rosa Blanes-Hernández, Julián de la Torre-Cisneros, Luis E. López-Cortés, Elena García de Vinuesa-Calvo, Clara M. Rosso, Jerónimo Pachón, Elisa Cordero.

**Writing – original draft:** Sonsoles Salto-Alejandre.

**Writing – review & editing:** Nuria Sabé, Antonio Ramos-Martínez, Laura Linares, Maricela Valerio, Pilar Martín-Dávila, Mario Fernández-Ruiz, María Carmen Fariñas, Marino Blanes-Juliá, Elisa Vidal, Zaira R. Palacios-Baena, Román Hernández-Gallego, Jordi Carratalá, Jorge Calderón-Parra, María Ángeles Marcos, Patricia Muñoz, Jesús Fortún-Abete, José María Aguado, Francisco Arnaiz-Revillas, Rosa Blanes-Hernández, Julián de la Torre-Cisneros, Luis E. López-Cortés, Elena García de Vinuesa-Calvo, Clara M. Rosso, Jerónimo Pachón, Javier Sánchez-Céspedes, Elisa Cordero.

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
