## [Decision Letter · Decision Letter 0]

8 Feb 2021

PONE-D-21-01017

Risk factors for unfavorable outcome and impact of early post-transplant infection in solid organ recipients with COVID-19: A prospective multicenter cohort study

PLOS ONE

Dear Dr. Pachon,

Thank you for submitting your manuscript to PLOS ONE. After careful consideration, we feel that it has merit but does not fully meet PLOS ONE’s publication criteria as it currently stands. Therefore, we invite you to submit a revised version of the manuscript that addresses the points raised during the review process.

The main criticism is the quality of writing. The authors need to re-edit the entire manuscript for English and grammer.

The manuscript needs to be revised as indicated by the reviewers:

Reviewer # 1:

The manuscript is well written. I have one question: in the analysis, there were 10 patients that were discharged and readmitted during the study period. For those 10 patients, how many days between their discharge and readmission? How many of them yield unfavorable outcomes? Will the results change significantly if including their 2nd hospital admission episode in the logistic regression? Because the p-value for lactate dehydrogenase >= 300 is just slightly smaller than 0.05 and its 95% CI is also very close to 1, it is possible that a slight change in the data would change the conclusion. The same for Lymphocytes < 1, p-value close to 0.05 and 95% CI close to 1.

Another question is: some variables were “dichotomized based on normal ranges” in the analysis. I would like to know if these thresholds were determined by authors or based on some well-established or well-accepted criteria.

Reviewer # 2

This is an interesting paper. The study is well designed and with proper statistical analysis of the data.  However, the manuscript is poorly written and needs extensive editing to correct for typos and grammatical errors. I have few minor comments.

Line 159: Replace  confusion by confounding

Line 311. Clarify hypothetically

Line 327: Spell out the “abovementioned unfavorable outcomes”

Line 329: What is severity risk factor?

Line 349:   Replace “cardiopathy”  with  “cardiomyopathy”

Line 351: Add “and” after “included, “

Line 377: Replace “proved” with “showed”.

Line 385: Replace “accurate” with “strong

We look forward to receiving your revised manuscript.

Kind regards,

Stanislaw Stepkowski

Academic Editor

PLOS ONE

Journal Requirements:

2. In your Methods section, please provide additional information about the participant recruitment method and the demographic details of your participants.

Please ensure you have provided sufficient details to replicate the analyses such as:

a) a description of any inclusion/exclusion criteria that were applied to participant recruitment, and

b) a statement as to whether your sample can be considered representative of a larger population.

5. One of the noted authors is a group; The COVIDSOT Working Team.

In addition to naming the author group, please list the individual authors and affiliations within this group in the acknowledgments section of your manuscript.

Please also indicate clearly a lead author for this group along with a contact email address.

Reviewers' comments:

Reviewer's Responses to Questions

**Comments to the Author**

1. Is the manuscript technically sound, and do the data support the conclusions?

Reviewer #1: Yes

Reviewer #2: Yes

2. Has the statistical analysis been performed appropriately and rigorously? 

Reviewer #1: Yes

Reviewer #2: Yes

3. Have the authors made all data underlying the findings in their manuscript fully available?

Reviewer #1: Yes

Reviewer #2: Yes

4. Is the manuscript presented in an intelligible fashion and written in standard English?

Reviewer #1: Yes

Reviewer #2: No

5. Review Comments to the Author

Reviewer #1: The manuscript is well written. I have one question: in the analysis, there were 10 patients that were discharged and readmitted during the study period. For those 10 patients, how many days between their discharge and readmission? How many of them yield unfavorable outcomes? Will the results change significantly if including their 2nd hospital admission episode in the logistic regression? Because the p-value for lactate dehydrogenase >= 300 is just slightly smaller than 0.05 and its 95% CI is also very close to 1, it is possible that a slight change in the data would change the conclusion. The same for Lymphocytes < 1, p-value close to 0.05 and 95% CI close to 1.

Another question is: some variables were “dichotomized based on normal ranges” in the analysis. I would like to know if these thresholds were determined by authors or based on some well-established or well-accepted criteria.

Reviewer #2: This is an interesting paper. The study is well designed and with proper statistical analysis of the data. However, the manuscript is poorly written and needs extensive editing to correct for typos and grammatical errors. I have few minor comments.

Line 159: Replace confusion by confounding

Line 311. Clarify hypothetically

Line 327: Spell out the “abovementioned unfavorable outcomes”

Line 329: What is severity risk factor?

Line 349: Replace “cardiopathy” with “cardiomyopathy”

Line 351: Add “and” after “included, “

Line 377: Replace “proved” with “showed”.

Line 385: Replace “accurate” with “strong”

6. PLOS authors have the option to publish the peer review history of their article (what does this mean?). If published, this will include your full peer review and any attached files.

Reviewer #1: No

Reviewer #2: **Yes: **sadik A. khuder

---

## [Author Response · Author response to Decision Letter 0]

10 Mar 2021

March 10, 2021

Stanislaw Stepkowski

Academic Editor

PLOS ONE

Dear Dr Stepkowski,

We are sending the revised version of our manuscript “Risk factors for unfavorable outcome and impact of early post-transplant infection in solid organ recipients with COVID-19: A prospective multicenter cohort study (PONE-D-21-01017)”. In this revised version we have addressed all the questions and the requests of the Reviewers, which are detailed below.

Moreover, to assure the quality of writing we have made a proofreading process to re-edit the entire manuscript for English and grammar.

Regarding the Journal additional requirements, we have: 1. assured to meet the PLOS ONE´s requirements; 2. included in the Methods section the information about the participants recruitment and demographics (line 123); 3. uploaded the minimal anonymized data set as Supporting Information files (S7 File); 4. added the ORCID iD of the corresponding author in Editorial Manager; and 5. added the information on the COVIDSOT Working Team in the acknowledgments section.

We hope that this revised version of the manuscript will fulfill the requirements to be accepted for publication in PLOS ONE.

Kind regards,

Jerónimo Pachón, MD, PhD

Emeritus Professor of Medicine

Institute of Biomedicine of Seville, Virgen del Rocio University Hospital, University of Seville

Av. Manuel Siurot s/n, 41013, Seville, Spain

Email: pachon@us.es

ORCID: 0000-0002-8166-5308

 

Reviewer # 1:

The manuscript is well written. I have one question: in the analysis, there were 10 patients that were discharged and readmitted during the study period. For those 10 patients, how many days between their discharge and readmission? How many of them yield unfavorable outcomes? Will the results change significantly if including their 2nd hospital admission episode in the logistic regression? Because the p-value for lactate dehydrogenase >= 300 is just slightly smaller than 0.05 and its 95% CI is also very close to 1, it is possible that a slight change in the data would change the conclusion. The same for Lymphocytes < 1, p-value close to 0.05 and 95% CI close to 1.

Thanks to the Reviewer by his positive appreciation of the manuscript. Please see below the answers to the two questions.

Regarding the question on the 10 patients readmitted during the study period: i) the median of days from discharges to readmissions was 7 days (range 1 to 19); therefore, we consider the readmission as part of the same episode of SARS-CoV-2 infection; ii) only one patient needed intensive care after readmission, but he also required it during his first admission; and the 10 patients were discharged alive after the readmissions; iii) in summary, there was no change in the number of unfavorable outcome (a composite of ICU admission and/or death) in the cohort of the 210 solid organ transplantation recipients. 

Another question is: some variables were “dichotomized based on normal ranges” in the analysis. I would like to know if these thresholds were determined by authors or based on some well-established or well-accepted criteria.

To examine factors associated with unfavorable clinical outcome, we dichotomized quantitative variables based on normal ranges, and after addressing their effect as continuous variables. The cut-offs used to dichotomize the variables were chosen based both in the normal values and in the cut-offs associated with unfavorable outcome when we analyzed the general population (Salto-Alejandre S et al. J Infect 2021 Feb;82(2):e11-e15. doi: 10.1016/j.jinf.2020.09.023. Epub 2020 Sep 25). 

Following the Reviewer question, to clarify it we have changed the sentence of the previous manuscript as follows: “… quantitative variables were dichotomized based on normal ranges and in the cut-offs associated with unfavorable outcome in the general population [28], after addressing their effect as continuous.” (lines 155-157 of the revised manuscript). We have added this new reference [28] in the manuscript.

Reviewer # 2

This is an interesting paper. The study is well designed and with proper statistical analysis of the data. However, the manuscript is poorly written and needs extensive editing to correct for typos and grammatical errors. I have few minor comments.

Thanks to the Reviewer by his positive appreciation of the design and the data analysis.

Line 159: Replace confusion by confounding

We have replaced confusion by confounding (line 159 of the revised manuscript), as requested by the Reviewer.

Line 311. Clarify hypothetically

Besides the main objective of the study, we wanted to supply descriptive data on patients with possible poorer outcome, because of graft dysfunction (12 patients) or developing COVID-19 in the period of maximal immunosuppressive therapy as it is the first month after receiving the transplant (6 patients). In this regard “hypothetically” was used as synonymous of “possible”. To clarify it, we have substituted the sentence “Two subgroups of the study population were considered, hypothetically, of higher risk” for “Two subgroups of the study population were considered of possible higher risk” (line 309 of the revised manuscript). 

Line 327: Spell out the “abovementioned unfavorable outcomes”

Following the Reviewer request we have changed it to “ICU admission and/or death” (lines 318-319 of the revised manuscript).

Line 329: What is severity risk factor?

Severity was used as synonymous of unfavorable outcome. To clarify it, we have changed this sentence as follows: “… as a risk factor for unfavorable outcome.” (lines 319-320 of the revised manuscript).

Line 349: Replace “cardiopathy” with “cardiomyopathy”

Following this request, we have replaced “cardiopathy” with “cardiomyopathy” (line 340 of the revised manuscript).

Line 351: Add “and” after “included, “

As requested, we have added “and” (line 342 of the revised manuscript).

Line 377: Replace “proved” with “showed”.

As requested, we have replaced “proved” with “showed” (line 368 of the revised manuscript).

Line 385: Replace “accurate” with “strong”.

As requested, we have replaced “accurate” with “strong” (line 376 of the revised manuscript).

---

## [Decision Letter · Decision Letter 1]

14 Apr 2021

Risk factors for unfavorable outcome and impact of early post-transplant infection in solid organ recipients with COVID-19: A prospective multicenter cohort study

PONE-D-21-01017R1

Dear Dr. Pachon,

We’re pleased to inform you that your manuscript has been judged scientifically suitable for publication and will be formally accepted for publication once it meets all outstanding technical requirements.

Kind regards,

Stanislaw Stepkowski

Academic Editor

PLOS ONE

Additional Editor Comments (optional):

None

Reviewers' comments:

Reviewer's Responses to Questions

**Comments to the Author**

1. If the authors have adequately addressed your comments raised in a previous round of review and you feel that this manuscript is now acceptable for publication, you may indicate that here to bypass the “Comments to the Author” section, enter your conflict of interest statement in the “Confidential to Editor” section, and submit your "Accept" recommendation.

Reviewer #2: All comments have been addressed

2. Is the manuscript technically sound, and do the data support the conclusions?

Reviewer #2: Yes

3. Has the statistical analysis been performed appropriately and rigorously? 

Reviewer #2: Yes

4. Have the authors made all data underlying the findings in their manuscript fully available?

Reviewer #2: Yes

5. Is the manuscript presented in an intelligible fashion and written in standard English?

Reviewer #2: Yes

6. Review Comments to the Author

Reviewer #2: The authors have addressed all of my comments appropriately. I recommend acceptance of this manuscript.

7. PLOS authors have the option to publish the peer review history of their article (what does this mean?). If published, this will include your full peer review and any attached files.

Reviewer #2: **Yes: **Sadik Khuder

---

## [Editor Report · Acceptance letter]

19 Apr 2021

PONE-D-21-01017R1 

Risk factors for unfavorable outcome and impact of early post-transplant infection in solid organ recipients with COVID-19: A prospective multicenter cohort study 

Dear Dr. Pachón:

I'm pleased to inform you that your manuscript has been deemed suitable for publication in PLOS ONE. Congratulations! Your manuscript is now with our production department. 

Kind regards, 

on behalf of

Dr. Stanislaw Stepkowski 

Academic Editor

PLOS ONE